# Designing Stable *Bacillus anthracis* Antigens with a View to Recombinant Anthrax Vaccine Development

**DOI:** 10.3390/pharmaceutics14040806

**Published:** 2022-04-06

**Authors:** Ekaterina M. Ryabchevskaya, Dmitriy L. Granovskiy, Ekaterina A. Evtushenko, Peter A. Ivanov, Olga A. Kondakova, Nikolai A. Nikitin, Olga V. Karpova

**Affiliations:** Department of Virology, Faculty of Biology, Lomonosov Moscow State University, 119234 Moscow, Russia; dgran98@gmail.com (D.L.G.); trifonova.katerina@gmail.com (E.A.E.); pivanov@belozersky.msu.ru (P.A.I.); olgakond1@yandex.ru (O.A.K.); nikitin@mail.bio.msu.ru (N.A.N.); okar@genebee.msu.ru (O.V.K.)

**Keywords:** *Bacillus anthracis*, recombinant protective antigen, vaccines, stabilisation, structurally modified plant viruses, tobacco mosaic virus

## Abstract

Anthrax is a disease caused by *Bacillus anthracis* that affects mammals, including humans. Recombinant *B. anthracis* protective antigen (rPA) is the most common basis for modern anthrax vaccine candidates. However, this protein is characterised by low stability due to proteolysis and deamidation. Here, for the first time, two modification variants leading to full-size rPA stabilisation have been implemented simultaneously, through deamidation-prone asparagine residues substitution and by inactivation of proteolysis sites. Obtained modified rPA (rPA83m) has been demonstrated to be stable in various temperature conditions. Additionally, rPA1+2 containing PA domains I and II and rPA3+4 containing domains III and IV, including the same modifications, have been shown to be stable as well. These antigens can serve as the basis for a vaccine, since the protective properties of PA can be attributed to individual PA domains. The stability of each of three modified anthrax antigens has been considerably improved in compositions with tobacco mosaic virus-based spherical particles (SPs). rPA1+2/rPA3+4/rPA83m in compositions with SPs have maintained their antigenic specificity even after 40 days of incubation at +37 °C. Considering previously proven adjuvant properties and safety of SPs, their compositions with rPA83m/rPA1+2/rPA3+4 in any combinations might be suitable as a basis for new-generation anthrax vaccines.

## 1. Introduction

Anthrax is a dangerous disease that affects mammals, including humans. It is caused by a gram-positive spore-forming bacterium, *Bacillus anthracis*. Modern anthrax history documents several accidents, including cases of bioterrorism-related inhalational anthrax infection in the USA in 2001 [1] and a 2016 outbreak of anthrax in Siberia [2]. Given the danger of bioterror attacks using *B. anthracis* spores and local anthrax outbreaks, the existence of an effective anthrax vaccine is required. At the same time, licensed anthrax vaccine formulations based on non-encapsulated *B. anthracis* strains cell filtrates—Anthrax Vaccine Adsorbed (AVA, Biothrax^TM^, Lansing, MI, USA) and Anthrax Vaccine Precipitated (AVP, UK), as well as live attenuated anthrax vaccines licensed in Russia and China, have some significant disadvantages. The use of these vaccines has been commonly associated with adverse reactions [3]. In cell filtrate-based vaccines, such reactions might have been due to the indeterminate quantity of anthrax toxin edema factor and lethal factor or of other bacterial proteins, while their contribution to protective immune response is under-researched [4]. Thus, the development of an effective new-generation recombinant vaccine is desirable.

Most modern anthrax vaccines and vaccine candidates target anthrax toxin. It consists of three proteins encoded by the large virulence plasmid pXO1: a protective antigen, a lethal factor and an edema factor. Protective antigen (PA, 83 kDa) is totally harmless to humans in the absence of other toxin components, because the PA’s only function is to transport the edema factor and lethal factor through the host cell’s barriers. What is crucial about this protein is that PA-induced antibodies are sufficient for the induction of a protective anti-anthrax immune response [5]. Therefore, the most convenient way to design a new vaccine formulation is to use recombinant PA (rPA). However, the stability of rPA is very low and decreases when rPA is adsorbed to aluminium hydroxide, the adjuvant used in AVA and in almost all candidate anthrax vaccines being in clinical trials [6,7,8]. The instability of rPA is due to spontaneous deamidation of asparagine residues and proteolysis at furin and chymotrypsin cleavage sites [9,10,11]. The effective stabilisation of anthrax protective antigen and the development of a proper adjuvant for it are problems that remain to be solved.

Since deamidation and proteolysis are the main reasons for rPA instability, the most obvious solution is a site-specific mutagenesis of deamidation-prone asparagine residues and inactivation of proteolysis sites. Among the 68 PA asparagine residues that are vulnerable to deamidation, there are several that are mostly relevant because of their half-lives and their impact on rPA properties [9,10]. It has been shown that the replacement of Asn^713^ and Asn^719^ by Gln leads to an increase in rPA stability. Modified protein has induced toxin-neutralising antibodies at a similar level to that elicited by fresh, non-modified protein. However, in contradistinction to non-modified rPA, modified protein kept the ability to elicit the same level of antibodies after storage [12]. Two major proteolysis-sensitive sites in the structure of PA are a furin-sensitive site in domain I and a chymotrypsin-sensitive site in domain II. Their inactivation has also been shown to lead to rPA stabilisation [11].

Individual isolated PA domains may have protective properties [13,14]. Therefore, the development of an anthrax vaccine might include the use of recombinant proteins containing different combinations of PA domains, rather than the full-size protein. This approach should ease the task of obtaining anthrax antigens. Previously, we have created a protein containing III and IV PA domains with Asn^713^ and Asn^719^ replaced by Gln (rPA3+4). This protein demonstrated a relatively high level of stability [15]. At the same time, there are data to suggest that the highest quantity of protective epitopes is located in the IV domain of the protein [13].

Specific rPA stabilisation provided by modifications of rPA structure can be improved by the addition of non-specific stabilisation through the formation of compositions with carrying platforms. Previously, we have shown that recombinant full-size PA (rPA83) in composition with spherical particles (SPs) obtained from the tobacco mosaic virus, through its thermally induced structural rearrangement, is more stable than when it is in the form of an individual protein [15]. The use of plant viruses or their virus-like particles as carrying platforms for different antigens is a fairly common concept in vaccine development [16,17]. SPs have been proven to have immunostimulating properties [17], while being biodegradable and safe for mammals [18,19,20]. Considering the foregoing discussion, SPs could be a promising carrying platform and stabiliser for rPA83.

Even though we consider rPA3+4 as a prospective vaccine basis due to the presence of a neutralising epitope-rich domain IV, the development of a stable vaccine containing all PA domains is still an objective to be achieved. In the current study, a recombinant protein that contained I and II PA domains with furin-sensitive and chymotrypsin-sensitive sites mutations, which led to their deactivation, (rPA1+2) was designed and obtained. The study also featured the construction and production of a full-size modified PA containing each structural modification implemented in rPA3+4 and rPA1+2 (rPA83m). The stability of rPA3+4, rPA1+2 and rPA83m was evaluated and was found to be relatively high. Compositions of any one of obtained proteins with SPs and compositions of rPA1+2 and rPA3+4 simultaneously adsorbed to SPs were obtained and the influence of SPs on the antigens’ stability was evaluated. Current data show that all three designed modified recombinant proteins are hypothetically able to serve as a basis for anthrax vaccine development and that SPs are a promising carrying platform for these antigens.

## 2. Materials and Methods

### 2.1. Obtaining Genetic Constructs

*B. anthracis* PA amino acid sequence [21] str.A2012 was taken as the basis for designing recombinant proteins. rPA1+2 consists of 485 amino acids corresponding to residues 1–487 of full-length PA Uniprot sequence Q08G54 with the substitution of ^162^NSRKKR^167^ with ^162^QSSNKE^167^ and the deletion of two phenylalanines (F) in 313 and 314 positions. The modified amino acid sequence was subjected to reverse translation in silico and optimised for expression in an *E. coli* bacterial system as described previously [15]. An optimised variant of synthetic rPA1+2 gene was assembled from overlapping oligonucleotides by the Evrogen company (Moscow, Russia) and cloned into a pQE-30 expression vector (Qiagen, Hilden, Germany) between the BamHI and HindIII cleavage sites. The resulting construct was named pQE-30-rPA1+2. In pQE-30 vector BamHI follows the hexahistidine-tag coding sequence, which is in the same coding frame. Thus, pQE-30 provides the expression of cloned proteins with N-ter3m his-tag for further metal affinity purification.

The genetic construct for rPA83m was generated using pQE-30-rPA1+2 and a previously obtained rPA3+4 expression vector (pQE-30-rPA3+4) [15] as templates. During the first stage, two DNA fragments corresponding to 3′-fragment of rPA1+2 gene (PCR1) and to 5′-fragment of rPA3+4 gene (PCR2) were obtained using primers presented in Table 1. PCR2 also contained 20 nucleotides, which were specific to the 3′-end of the rPA1+2-coding sequence. At the second stage, overlap PCR with PCR1 and PCR2 was conducted, and SphIFwd and PstIRev flanking primers were used for amplifying the fusion product (PCR3). PCR3 was digested with SphI and PstI. The 3′-fragment of the rPA3+4 gene was obtained by restricting the pQE-30-rPA3+4 vector with PstI and HindIII. The pQE-30-rPA1+2 vector digested with SphI and HindIII was employed for the one-step cloning of PCR3 and the 3′-fragment of rPA3+4. The resulting construct was named pQE-30-rPA83m. A graphical overview of the generation of the rPA83m expression vector is presented in Figure 1.

### 2.2. Protein Expression and Purification

*E*. *coli* strain SG13009 was used for the expression of recombinant proteins. The over-night cultures were grown in 3 mL 2YT medium (1.6% (*w*/*v*) tryptone, 1% (*w*/*v*) yeast extract, 0.5% (*w*/*v*) NaCl) with 50 µg/mL of kanamycin and 100 µg/mL of ampicillin at 37 °C, with shaking at 180 rpm. These cultures were transferred into 200 mL 2YT with the same content of antibiotics and were cultivated at +37 °C, with shaking at 180 rpm for 3 h; they were then induced with 2 mM IPTG. For expression of rPA3+4 protein cultures were grown for 4 h at +37 °C and 180 rpm. For expression of rPA1+2 and rPA83m proteins, cultures were grown for 4–6 h at +20 °C and 180 rpm. Following induction, cell pellets were harvested by centrifugation for 10 min at 5000× *g* and stored at −20 °C until use. Cell pellets were lysed in 6 M GuHCl. 0.2% (*w*/*v*) natrium deoxycholate was added to GuHCl lysis buffer in order to eliminate endotoxins. Since proteins’ genetic constructs were pQE-30 based, all accumulated recombinant proteins contained hexahistidine-tag at the N-term and were purified by metal affinity chromatography with Ni^2+^-NTA resin (Qiagen, Hilden, Germany) under denaturing conditions with 6 M GuHCl and 8 M urea, according to the manufacturer’s protocol. Proteins eluted from the column were dialysed against deionised water using 12–14 kDa MWCO dialysis tubes (Servapor, Heidelberg, Germany), in the ratio 1/250, for 4 h, replacing water hourly and stored as water solution at −20 °C.

### 2.3. Western Blot

Proteins were separated by electrophoresis in an 8–20% SDS-PAGE and transferred to PVDF membrane (Invitrogen, TM, ThermoFisher Scientific, Waltham, MA, USA) using a Pierce™ Power Blotter transfer system (Thermo Fisher Scientific, Waltham, MA, USA). The membrane was blocked with 5% (*w*/*v*) non-fat dry milk in TTBS (0.01 M Tris-HCl (pH 7.4), 0.15 M NaCl, 0.05% (*v*/*v*) Tween-20) and treated with polyclonal mouse antiserum to non-modified rPA83 or chimeric recombinant antibodies to PA in 1:5000 dilution. Polyclonal mouse antiserum to non-modified rPA83 had been obtained previously in our laboratory [15] and chimeric recombinant antibodies to PA had been previously obtained [22] and generously provided by our colleagues Dr. Teimur K. Aliev and Dr. Anna A. Panina from Shemyakin-Ovchinnikov Institute of bioorganic chemistry. Secondary HRP-conjugated antibodies to mouse IgG (115-035-003, Jackson ImmunoResearch Laboratories, Inc., West Grove, PA, USA) or to human IgG (1G1cc, HyTest, Moscow, Russia) were used in 1:10,000 dilutions. The WesternBright ECL chemiluminescent substrate (Advansta Inc., San Jose, CA, USA) was used and the signal was detected using the ChemiDoc XRS+ gel documentation system (Bio-Rad Laboratories, Inc., Hercules, CA, USA).

### 2.4. Obtaining SPs

Tobacco mosaic virus with a concentration of 2 mg/mL was used for obtaining SPs according to the protocol described in Trifonova et al. (2015) [23].

### 2.5. Labelling of Proteins and Fluorescence Microscopy

In this case, rPA1+2 and rPA3+4 were labelled with fluorescein isothiocyanate (FITC) or rhodamine isothiocyanate (RITC), respectively. 50 µL of freshly prepared FITC (Sigma, St. Louis, MO, USA) or RITC (Sigma, St. Louis, MO, USA) in anhydrous DMSO (1 mg/mL) was slowly added, in aliquots, with gentle stirring, to 500 µL of corresponding protein solution in carbonate-bicarbonate buffer (pH = 9.1). rPA1+2^FITC^ and rPA3+4^RITC^ solutions were incubated overnight at +4 °C in the dark and then dialysed in the dark against 1 × PBS in the ratio 1/2000 for 4 h, replacing the 1 × PBS every hour. For fluorescence analysis, samples were loaded on formvar-coated coverslips and dried in air in the dark. Before checking, coverslips were treated with a photo-protector 1,4-diazabicyclo[2.2.2]octane and fixed on the slide, followed by examination under an Axiovert 200M fluorescence microscope (Carl Zeiss, Gottingen, Germany) equipped with an ORCAII-ERG2 integrated camera (Hamamatsu Photonics, Hamamatsu City, Japan).

### 2.6. Immunofluorescence Analysis

Immunofluorescence analyses of SPs-rPA1+2, SPs-rPA3+4 and SPs-rPA83m compositions were performed as described previously [24]. Primary antibodies were used in 1/100 dilution; secondary anti-mouse IgG antibodies conjugated to Alexa Fluor^®^ 546 (A-11030, Invitrogen, Thermo Fisher Scientific, Waltham, MA, USA) or secondary anti-human IgG antibodies conjugated to CF™ 488A (SAB4600055, Sigma-Aldrich, St. Louis, MO, USA) were used in 1/200 or 1/100 dilutions, respectively.

### 2.7. Protein Stability Analysis

A modified recombinant anthrax antigen stability analysis was performed by protein electrophoresis in 8–20% SDS-PAGE with staining by Coomassie G-250. 1 µg of antigen was analysed in each gel line.

### 2.8. 2D-Electrophoresis Analysis

2D-electrophoresis with the following Coomassie G-250 staining was performed as described in Zvereva et al., 2015 [25] (second variant). Ampholines with pH 4.6–10 were used. Equal amount (10 µg) of the rPA83m incubated at +4 °C or at +37 °C was loaded to the “acid border” of gel column.

For 2D-electrophoresis with fluorescence detection equal amount (3 µg) of the rPA83m incubated at +4 °C or at +37 °C was labelled with 400 pmol of Cyanine2 (cat. #3A041) or Cyanine5 (cat. #3C041) NHS ester minimal dyes (Lumiprobe RUS Ltd., Moscow, Russia), respectively, according to the manufacturer’s instructions. After samples were mixed, 0.75% (*w*/*v*) DTT and 0.4% (*v*/*v*) Ampholine 3–10 (Bio-Rad Laboratories, Hercules, CA, USA) were added. IEF was performed using 3-10NL nonlinear ReadyStrip IPG Strips (cat. #1632009, Bio-Rad Laboratories, Hercules, CA, USA). Isoelectrofocusing (IEF) was performed using PROTEAN i12 IEF system (Bio-Rad Laboratories, Hercules, CA, USA). After IEF, the ejected strips were incubated in equilibration buffer (6 M urea, 0.375 M Tris-HCl pH 8.8, 40% (*w*/*v*) glycerol, 2% (*w*/*v*) SDS, 2% DTT (*w*/*v*)) for 30–45 min. The tube gels were placed onto polyacrylamide gels (12%) of 1-mm thickness, 20 × 20 cm (EV-20, Helicon, Moscow, Russia) and fixed using 1.0% (*w*/*v*) agarose containing 0.01% (*w*/*v*) bromphenol blue. The electrophoresis was carried out at 120–200 V for 5–7 hours. The gels were scanned with a TYPHOON FLA950 (GE Healthcare, Chicago, IL, USA). The 100 μm resolution of the scanner was used. The 473 nm laser (with the voltage adjusted to 800 V) or the 635 nm laser (with the voltage adjusted to 800 V) was used for detection of Cyanine2 or Cyanine5, respectively. DBR1 (530DF20/665LP) emission filter was applied in both cases of Cyanine2 (515–545 nm wavelength) and Cyanine5 (>665 nm wavelength) detection.

## 3. Results

### 3.1. Designing and Obtaining Stable Recombinant Anthrax Antigens

In order to obtain stable recombinant antigens of *B. anthracis*, three genetically modified recombinant proteins, rPA3+4, rPA1+2 and rPA83m, were designed, obtained and analysed. Native PA consists of four domains (Figure 2a). The genetic construction of rPA3+4 containing III and IV domains of PA with Asn^713^ and Asn^719^ substitutions for glutamines (Figure 2c) had been generated previously [15]. Here, rPA1+2 containing domains I and II of PA with proteolytic sites modifications leading to their deactivation was also developed. Based on results obtained by Ramirez et al. (2002) [11], in order to inactivate proteolysis sites, the sequence of ^164^RKKR^167^ was replaced by ^164^SNKE^167^ at the furin cleavage site and two phenylalanines in the positions 313 and 314 were deleted at the chymotrypsin cleavage site (Figure 2b). Deamidation-prone Asn^162^ [9] was also substituted for glutamine within the rPA1+2 sequence. The construction of rPA83m (Figure 2d) was obtained on the basis of rPA1+2 and rPA3+4 expression vectors by overlap PCR. Therefore, rPA83m included all modifications implemented in rPA1+2 and rPA3+4. A graphical overview of recombinant antigens, with each modification introduced highlighted in comparison to the native PA, is shown in Figure 2.

The molecular weights of rPA3+4, rPA1+2 and rPA83m, calculated by amino acid sequence via the ProtParam EXPaSy proteomics server (Swiss Institute of Bioinformatics, http://expasy.org/ (accessed on 24 February 2022), were 30 kDa, 56 kDa and 84 kDa, respectively; these are consistent with those revealed by SDS-PAGE analyses (Figure 3a, lanes 3, 2 and 1). Modified antigens were shown to be recognised by polyclonal antiserum to non-modified recombinant full-size PA (Figure 3b, lanes 3, 2 and 1). The rPA83m and rPA3+4 were also shown to interact clearly with recombinant chimeric monoclonal anti-PA antibodies, which display neutralising activity [22] (Figure 3c, lanes 1 and 3). The recognition of proteins with higher electrophoretic mobility in recombinant protein samples by both polyclonal antiserum and monoclonal antibodies (Figure 3a–c, lanes 1–3, 5), along with the absence of a reaction with *E. coli* cell lysate (negative control) (Figure 3a–c, lane 6), proved these proteins to be antigenically active rPA fragments rather than contaminations from *E.coli* cells. Moreover, in the positive control sample with non-modified recombinant PA63 (rPA63), degraded forms of protein (Figure 3a, lane 5) prevailed. Conversely, in each of the modified rPA samples (Figure 3a, lanes 1, 2 and 3), full-size forms predominated, indicating their relatively high stability.

Since a solution to the PA degradation problem was the main aim of protein modifications, the stability of each modified recombinant anthrax antigen was evaluated. For this purpose, proteins were incubated under various temperature conditions. Before incubation and for each control point after incubation, samples containing an equal amount of protein were analysed. rPA83m remained stable after 144 days of incubation at +4 °C (Figure 4a, lanes 3–6), the appearance of proteins with higher electrophoretic mobility than rPA83m was not clearly detected and no decrease in rPA83m content in the samples was observed after incubation. On the contrary, non-modified rPA83 incubated at the same temperature underwent relatively rapid degradation and the full-size form was not detected at all after 33 days of incubation; furthermore, the set of proteins with higher electrophoretic mobility than rPA83 changed in samples during the incubation (Figure 4b, lines 1–7).

Thus, modifications implemented in rPA83m resulted in the considerably higher stability of the protein compared to the non-modified rPA83 at +4 °C, which is a common vaccine storage temperature.

In this case, rPA3+4 has previously been shown to be stable at +25 °C for at least 17 days [15]. The current study continued the assessment of its stability and demonstrated that rPA3+4 mostly remains stable, even after 160 days of incubation at +25 °C (Figure 4c, lanes 2, 3). Here, rPA1+2 was shown to have similar stability: after 160 days of incubation at +25 °C, the major band revealed by electrophoresis analysis (Figure 4c, lane 8) corresponded to the non-degraded protein. For rPA3+4, stability for 21 days at +37 °C, which was used to simulate the accelerated ageing of protein, was also demonstrated (Figure 4c, lane 5). After 160 days of rPA3+4 (Figure 4c, lane 6) and rPA1+2 (Figure 4c, lane 9) incubation at +37 °C, the bands corresponding to non-degraded protein were undetectable.

### 3.2. Modified Recombinant Anthrax Antigens’ Adsorption to SPs

In previous research, the authors have shown that spherical particles obtained from the tobacco mosaic virus are able to stabilise non-modified rPA83 [15]. In the current study, the examination of the adsorption of each of three modified anthrax antigens to SPs was performed using immunofluorescence analysis (Figure 5, Appendix A). It was shown that rPA83m (Figure 5a,b), rPA1+2 (Figure 5c,d) and rPA3+4 (Figure 5e,f) were able to form SPs-antigen compositions. According to the authors’ previous study, the SPs/antigen mass ratio was chosen to be 10/1 [17]. All compositions were obtained in PBS, which is an acceptable buffer for vaccine formulations. For fluorescence visualisation, SPs-antigen compositions were treated with mouse polyclonal antiserum to an appropriate antigen and with Alexa Fluor^®^ 546-conjugated secondary antibodies. In each case, the presence of a signal indicated the maintenance of antigenic properties of a corresponding modified anthrax antigen while being adsorbed to SPs. Moreover, the possibility of recognition by monoclonal neutralising antibodies was studied for the compositions of SPs-rPA83m (Figure 5g,h) and SPs-rPA3+4 (Figure 5i,j). CF™ 488A-conjugated secondary antibodies specific to human IgG were employed for visualisation. The presence of the green fluorescent signal corresponding to SPs’ location illustrated that a neutralising epitope was exposed and was available for interaction with antibodies within the compositions obtained. rPA1+2 and rPA3+4 simultaneously adsorbed to SPs could provide an alternative approach to vaccine development without using the full-size rPA.

In order to examine the possibility of simultaneous adsorption, rPA1+2 and rPA3+4 were labelled with fluorescein isothiocyanate (FITC) or rhodamine isothiocyanate (RITC), respectively and analysed by both fluorescence and SDS-PAGE analysis (Figure 6, Appendix A). The resulting labelled proteins were named rPA1+2^FITC^ (Figure 6d,e, lane 2) and rPA3+4^RITC^ (Figure 6d,e, lane 1). Compositions of SPs with both rPA1+2^FITC^ and rPA3+4^RITC^ (Figure 6d,e, lane 3) were obtained in PBS with the following mass ratio of the respective ingredients: 20/1/1, which corresponded to 10/1 of the SPs/total antigen ratio. Pictures obtained using fluorescence imaging mode in FITC (Figure 6a) and RITC (Figure 6b) channels and phase contrast imaging mode (Figure 6c) demonstrated that particles’ outlines in all three images totally correlated. Therefore, rPA1+2 and rPA3+4 were shown to be able to simultaneously cover the surface of SPs.

### 3.3. Evaluating the Stability of Modified Recombinant Anthrax Antigens in Compositions with SPs

Previously, it has been shown that SPs can increase the stability of non-modified rPA83 [15]. In the current study, the ability of SPs to stabilise each of three obtained modified anthrax antigens was evaluated. Compositions of each of the modified proteins with SPs were formed at the SPs/antigens mass ratio of 10/1. Samples of each individual antigen, as well as samples of SPs-antigen compositions, were prepared in PBS and incubated at +37 °C for 40 days to simulate accelerated protein ageing. Equal amounts of antigens were analysed before incubation and for each control point during incubation.

The band corresponding to the non-degraded rPA83m was clearly detectable after 40 days of incubation at +37 °C in the sample of SPs-rPA83m compositions (Figure 7a, lane 13), while in the sample of individual rPA83m the full-size protein band was barely detectable after 40 days of incubation (Figure 7a, lane 14). Similar results were obtained for rPA3+4 and rPA1+2: in composition with SPs, both rPA3+4 and rPA1+2 were clearly detectable after 40 days of incubation (Figure 7b, lane 13 and Figure 6c, lane 13, respectively), while individual antigens were barely detectable after 35 days in the case of rPA3+4 (Figure 7b, lane 12) and after 20 days in the case of rPA1+2 (Figure 7c, lane 6). At each control point, the stability of each anthrax antigen within SPs-antigen compositions was higher than the stability of the corresponding antigen in an individual formulation (Figure 7). Replicates for each of three stability experiments (Figure 7a–c) are presented in Appendix B (Figure A1a–c, respectively). For both replicates the relative analysis of recombinant proteins’ degradation rate through the experiment is presented in Table A1. Considering all obtained data, the stabilisation of modified recombinant anthrax antigens by SPs was demonstrated. It is worth noting that SPs protein suffers dimerisation and trimerisation while being stored, starting from day 10 in each of three stability evaluation experiments.

The maintenance of antigenic specificity for each modified anthrax antigen after 40 days of incubation within SPs-protein compositions at +37 °C was examined using immunofluorescence analysis (Figure 8 and Appendix A), in a similar way to which the antigens’ adsorption to SPs was evaluated. rPA83m (Figure 8a,b), rPA1+2 (Figure 8c,d) and rPA3+4 (Figure 8e,f) were shown to be able to interact with a polyclonal antiserum to an appropriate antigen. The correlation between the outline of each spherical particle in pictures obtained using fluorescence imaging mode and phase contrast imaging mode demonstrated that, after 40 days of incubation at +37 °C, each modified antigen covering SPs remained stable and kept its antigenic properties unchanged. For SPs-rPA83m (Figure 8g,h) and SPs-rPA3+4 (Figure 8i,j) compositions, the maintenance of neutralising epitope availability for recognition by antibodies after 40 days of incubation at +37 °C was also confirmed.

## 4. Discussion

Currently used anthrax vaccines have several disadvantages. The development of recombinant anthrax vaccine is desirable. The main problem with such vaccine design is the instability of rPA [7]. In the current research, we offer two approaches to rPA stabilisation. The first combines two variants of stabilising protein modifications: inactivation of proteolysis cleavage sites and deamidation-prone asparagine residue substitutions. Here, we pioneeringly implemented these two modification variants simultaneously. The second approach is the use of unique carrying platform with adjuvant properties based on plant viruses. The SPs ability to additionally improve the stability of modified full-length rPA83m containing the mutations in both proteolysis and deamidation sites was also studied for the first time.

Proteolysis and deamidation are thought to be the main causes of rPA instability [9,10,11]. Deamidation mediates the decrease in the rPA pI and the appearance of multiple rPA isoforms with pI values less than the one of freshly prepared protein. For non-modified rPA heterogeneity in terms of pI, which considerably increases after storage, thermal treatment or freeze/thaw cycles, was showed in several studies [9,10,26]. At the same time, rPA contains two major proteolysis sites [11], which predispose high fragmentation rate of the protein. Thus, suppression of deamidation and proteolysis can lead to stable rPA generation. To achieve this aim, three proteins were designed: rPA1+2 (56 kDa) containing I and II PA domains with inactivated furin and chymotrypsin cleavage sites (Figure 2b), rPA3+4 (30 kDa) containing III and IV PA domains with two crucial deamidation-prone Asn residues being replaced by Gln (had been developed previously by the authors’ group [15]) (Figure 2c) and rPA83m (84 kDa)—a full-size rPA in which both types of the aforementioned modifications were implemented (Figure 2d). Modifications of protease-sensitive regions were carried out according to results obtained by Ramirez et al. (2002). They showed that the replacement of the ^164^RKKR^167^ sequence by ^164^SNKE^167^ at the furin-sensitive site and the deletion of 313 and 314 phenylalanine residues at the chymotrypsin-sensitive site resulted in the generation of proteolysis-resistant rPA, which showed an ability to be accumulated in high concentrations. rPA modified in this way was able to induce the production of antibodies with neutralising properties [11]. Asn residues substitutions were based on research by Verma and Burns (2018) [12]. In their study, it was demonstrated that the replacement of located in the receptor-binding large loop domain deamidation-prone Asn^713^ and Asn^719^ residues by glutamines did not lead to a decrease in immunogenic properties. Moreover, this replacement increased immunogenicity compared with wild-type rPA in stored formulations [12]. The decrease in wild-type rPA immunogenicity after storage may be due to the inability of deamidated rPA to interact with its receptors on the surface of antigen-presenting cells, while this interaction was shown to have resulted in an enhanced toxin-neutralising antibody response [27]. Deamidation-prone Asn^162^ with a predicted half-life of 38 days [9] was also replaced by Gln in the present study. In the current research, approaches to full-size rPA stabilisation by site-specific mutagenesis of deamidation-prone asparagine residues and by inactivation of proteolysis sites have been combined for the first time. The deamidation of rPA83m was studied through assessing its pI upon storage. Zomber et al. (2005) showed by means of isoelectrofocusing that non-modified rPA divided into the multiple acidic isoforms [9] after incubation at +25 °C for as little as 7 days. In present work 2D-electrophoresis of rPA83m did not reveal the appearance of any additional protein isoforms after 7 days incubation at +37 °C (Appendix C, Figure A2a,b). Furthermore, the exact match of pI was demonstrated for the samples stored for 5 days at +4 °C and at +37 °C rPA83m (Appendix C, Figure A2c–e). That confirms that rPA83m is not considerably susceptible to deamidation and enables to suppose that rPA1+2 and rPA3+4 are also deamidation-resistant since they are identical to corresponding rPA83m regions sequences. Modified proteins obtained, containing particular PA domains (rPA1+2 and rPA3+4), may be a prospective basis for a vaccine since the protective properties of PA can be attributed to individual PA domains. It was demonstrated that separated PA domain IV was able to fully protect A/J mice against 100 median lethal doses of STI spores [13]. At the same time, sera from C57BL/6 mice immunised with PA domain I displayed a considerable ability to neutralise lethal toxin [14]. All three modified antigens obtained demonstrated an interaction with a polyclonal anti-rPA83 antiserum. Moreover, rPA3+4 and rPA83m were shown to interact with chimeric monoclonal anti-PA antibodies. These antibodies have a neutralising effect [22] and results obtained in the current work have confirmed that the epitope corresponding to them is located in either III or IV PA domain, since these antibodies do not interact with rPA1+2 while interacting with rPA3+4. Taking all of the data collected into account, three of the proteins obtained can be considered to be promising recombinant anthrax antigens. Another problem associated with rPA instability alongside deamidation and proteolysis is its susceptibility to aggregation. rPA rapidly loses its biological activity under high temperature conditions, and aggregation is considered to be the main reason for its thermal inactivation [28]. Dissolved rPA has been shown to form large aggregates when exposed to the temperatures above 44 °C. These aggregates could easily be visually detected [29,30,31]. Immunisation of mice with the aggregated rPA led to a dramatic decrease of toxin-neutralising antibodies (TNA) titers in blood sera, in comparison with TNA titers elicited by freshly prepared rPA [32].

The stability of modified anthrax antigens was evaluated. rPA83m protein remained stable at +4 °C (a common vaccine storage temperature) for at least 144 days. For comparison, non-modified rPA83 was not detectable after 14 days of incubation at +4 °C. Two groups of amino-acid sequence modifications provided the only difference between the analysed antigens, thus leading to dramatic protein stabilisation. rPA1+2 and rPA3+4 demonstrated a high level of stability at +25 °C (room temperature) for at least 160 days. This indicates that vaccines containing these PA domains-based antigens could be suitable for use even after disruption to the vaccine cold chain. Moreover, modified antigens were not shown to form visually detectable aggregates while being stored even at +37 °C (Appendix D, Figure A3a,b), in contradistinction to previously demonstrated features of non-modified rPA [27]. Further SDS-PAGE analysis was carried out for rPA83m specifically. The amount of rPA83m in the supernatant of centrifuged stored samples and in corresponding non-centrifuged stored samples turned out to be equal (Appendix D, Figure A3c,d). This evidence demonstrates the absence of large aggregates even after storage at +37 °C for 20 days.

All three modified antigens were shown to be stable, however, a certain level of degradation was still detected. Thus, an additional approach to rPA stabilisation was desirable. Previously, rPA83 adsorption to spherical particles obtained from the tobacco mosaic virus via its thermally induced structural rearrangement has been shown to lead to an increase in rPA83 stability [15]. At the same time, SPs have been demonstrated to have immunostimulating properties [17,33,34] and therefore presumably could serve not only as a stabiliser, but also as an adjuvant, being a more successful alternative to aluminum-based adjuvants [7,8]. It should be mentioned that the possibility of rPA stabilisation by chemically synthesised amphiphilic polyanhydride nanoparticles in dry-powdered formulations has been shown [35]. Moreover, rPA stabilised in this way induced a neutralising antibody response against *B. anthracis* with cyclic di-guanosine monophosphate used as an adjuvant [36]. In the light of these facts, we consider the adsorption to carrying platforms as a prospective way to stabilise rPA. Here, an examination was made of the adsorption of each of three modified anthrax antigens obtained to SPs. It was demonstrated that rPA1+2, rPA3+4 and rPA83m were still recognisable by polyclonal antiserum to an appropriate antigen while having been adsorbed to SPs. In addition, SPs-rPA3+4 and SPs-rPA83m compositions were able to interact with the above-mentioned chimeric monoclonal PA-neutralising antibodies, which indicates that the neutralising epitope is exposed. It enables assuming the presence of at least some native conformation within recombinant proteins. The further studies of immunogenicity and protectivity are necessary to confirm the acceptability of designed proteins as *B. anthracis* antigens. Nevertheless, the fact that this epitope remains recognisable by PA-neutralising antibodies being a part of modified recombinant anthrax antigens adsorbed to SPs, gives hope that the corresponding SPs-antigen compositions will be able to induce toxin-neutralising antibodies. The results obtained provide new opportunities for the use of modified anthrax antigens as a vaccine basis. For example, formulations based on both rPA1+2 and rPA3+4 adsorbed to SPs can provide a prospective field of research into the use of certain PA domains in vaccines. The current research has also demonstrated the possibility of simultaneous adsorption of these proteins to SPs with a view to developing a binary anthrax vaccine containing all PA epitopes while not including a full-size rPA.

The effective adsorption of rPA83m, rPA1+2 and rPA3+4 to SPs while maintaining the antigenic specificity of modified anthrax antigens within SPs-antigen compositions, and the possibility of SPs to stabilise non-modified rPA83, have encouraged consideration of the influence of SPs on the stability of each of three modified antigens. Stability was evaluated at +37 °C. Temperature conditions for the experiment were selected for the purpose of simulating accelerated protein ageing. At each control point, a pairwise visual comparison showed that anthrax antigens were less susceptible to degradation within SPs-antigen compositions than as individual proteins. In the latter case, each modified anthrax antigen was barely detected after 20–40 days of incubation, while within SPs-antigen compositions each antigen mostly remained stable for at least 40 days, which means that SPs were able to markedly stabilize rPA83m, rPA3+4 and rPA1+2. Immunofluorescence analysis carried out to examine the state of the antigen adsorbed to SPs for 40 days demonstrated the maintenance of antigenic specificity of each of three anthrax antigens and the ability of rPA83m and rPA3+4 to interact with monoclonal PA-neutralising antibodies. The latter indicated that the epitope crucial for anthrax toxin neutralisation was still exposed and was not subject to degradation during a relatively long incubation at +37 °C. We suggest that the interactions of rPA83m, rPA1+2, rPA3+4 with SPs surface slow antigens aging. Previously, we have shown the presence of hydrophobic amino acids on SPs surface [37]. Presumably, hydrophobic interactions between SPs and rPA antigens preserve the proteins conformation during storage and enhance their stability.

## 5. Conclusions

Results obtained in the current research enable the claim that modified anthrax antigens rPA1+2, rPA3+4 and rPA83m are resistant to degradation during extended storage under various temperature conditions and can serve as the basis for an anthrax vaccine. SPs can be used as a platform-carrier in such vaccines, since they are able to adsorb each of three modified anthrax antigens and increase their stability, while maintaining the antigenic properties of these antigens. Previously demonstrated adjuvant properties of SPs [17,33,34] suggest that SPs can serve not only as a stabiliser, but also as an adjuvant within SPs-recombinant antigen compositions. Safe anthrax vaccines that demonstrate long-term stability of formulation are needed as an essential means of mass immunisation in emergency situations. Further examination of SPs-rPA83m, SPs-rPA1+2 and SPs-rPA3+4 compositions will probably show that modified antigens of lesser size that therefore are easier to produce could offer sufficient levels of protection. Thus, we consider the compositions of SPs with any one of modified anthrax antigen or with the combination of rPA1+2 and rPA3+4 as prospective anthrax vaccine candidates that are worthy of further study. The ability of all obtained anthrax antigens to induce protective immunity is undoubtedly the next step in the development of stable and effective modern anthrax vaccine.

## Figures and Tables

**Figure 1 pharmaceutics-14-00806-f001:**
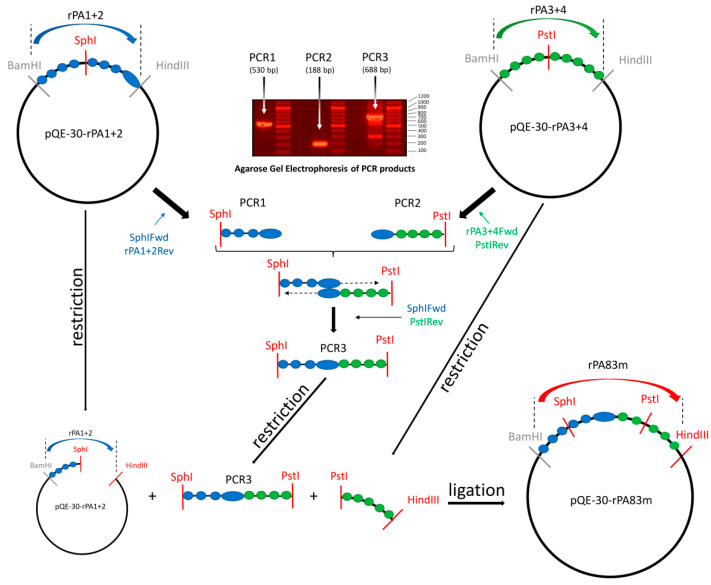
Graphical overview of the generation of the rPA83m expression vector. pQE-30-rPA1+2 and pQE-30-rPA3+4 are expression vectors containing nucleotide sequences of rPA1+2 and rPA3+4, respectively, that were cloned between the BamHI and HindIII cleavage sites. The rPA1+2-coding sequence is represented by blue rounds and the rPA3+4-coding sequence is represented by green rounds. The rPA1+2 3′-end sequence included in the rPA3+4Fwd primer is represented by the oval. Primers are indicated with the colour of the matrix to which they are specific. SphI and PstI restriction sites are located within the rPA1+2 and rPA3+4 nucleotide sequences, respectively. Restriction sites used for one-step cloning are marked in red. PCR1 is a product corresponding to the 3′-fragment of rPA1+2. PCR2 is a product corresponding to the 5′-fragment of rPA3+4 and also contains a 20-nucleotide length sequence that is specific to the 3′-end of rPA1+2. PCR3 is a fusion product generated by overlap PCR using PCR1 and PCR2 as primary fragments and SphIFwd and PstRev as flanking primers. 1% agarose gel electrophoresis (ethidium bromide staining) of purified PCR1 and PCR2, as well as a PCR mix containing amplified PCR3, is presented; the lengths of DNA ladder (GeneRuler DNA Ladder Mix, Thermo Scientific, Waltham, MA, USA) are indicated on the right.

**Figure 2 pharmaceutics-14-00806-f002:**
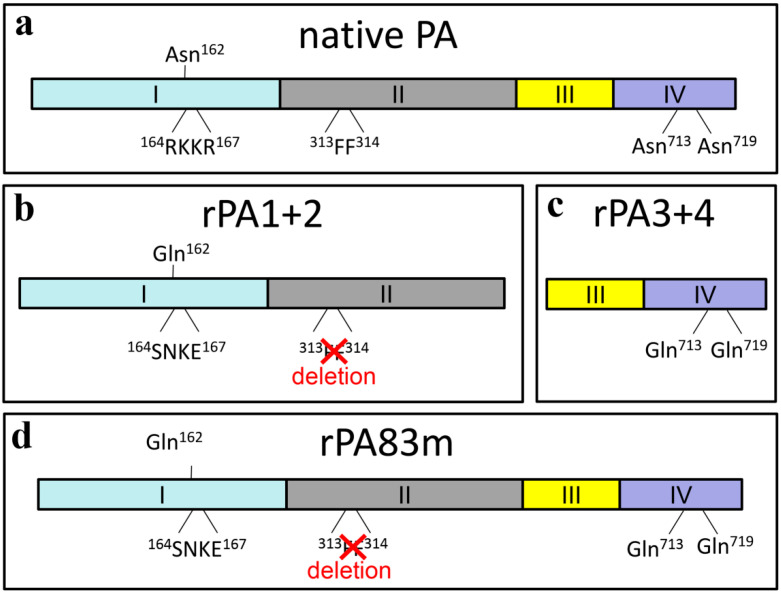
Graphical overview of modified recombinant anthrax antigens rPA1+2 (**b**), rPA3+4 (**c**) and rPA83m (**d**) compared with native PA (**a**). rPA1+2 contains PA domains I and II with mutations at furin and chymotrypsin cleavage sites (the ^162^NSRKKR^167^ fragment was replaced by ^162^QSSNKE^167^ at the furin proteolysis site and the ^313^FF^314^ fragment was deleted at the chymotrypsin proteolysis site). rPA3+4 contains PA domains III and IV with deamidation-prone Asn^713^ and Asn^719^ residues both replaced with Gln. rPA83m is a full-size modified recombinant PA with all modifications implemented in rPA1+2 and rPA3+4.

**Figure 3 pharmaceutics-14-00806-f003:**
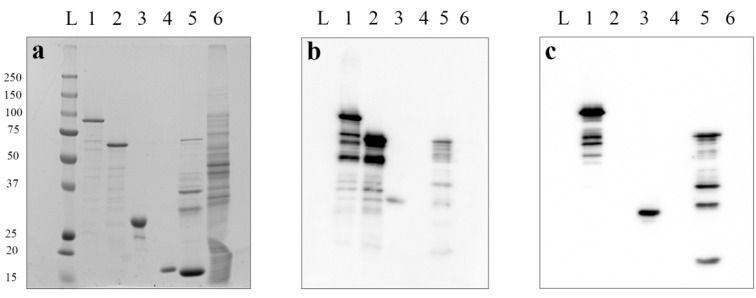
Interaction of obtained anthrax modified recombinant antigens with a polyclonal anti-rPA83 antiserum (**b**) and chimeric monoclonal PA-neutralising antibodies (**c**). L—protein molecular weight markers ladder (molecular weights, in kDa, are indicated on the left), 1—rPA83m, 2—rPA1+2, 3—rPA3+4, 4—Tobacco mosaic virus, 5—rPA63, 6—*E. coli* cell lysate. Electrophoresis analysis in 8–20% SDS-PAGE, staining by Coomassie G-250 (**a**). Western blot analysis with primary polyclonal antiserum to the non-modified rPA83 (1:5000) (**b**) or chimeric monoclonal PA-neutralising antibodies (1:5000) (**c**) and secondary HRP-conjugated antibodies (1:10,000). Original blot membranes’ images are presented in the Appendix A.

**Figure 4 pharmaceutics-14-00806-f004:**
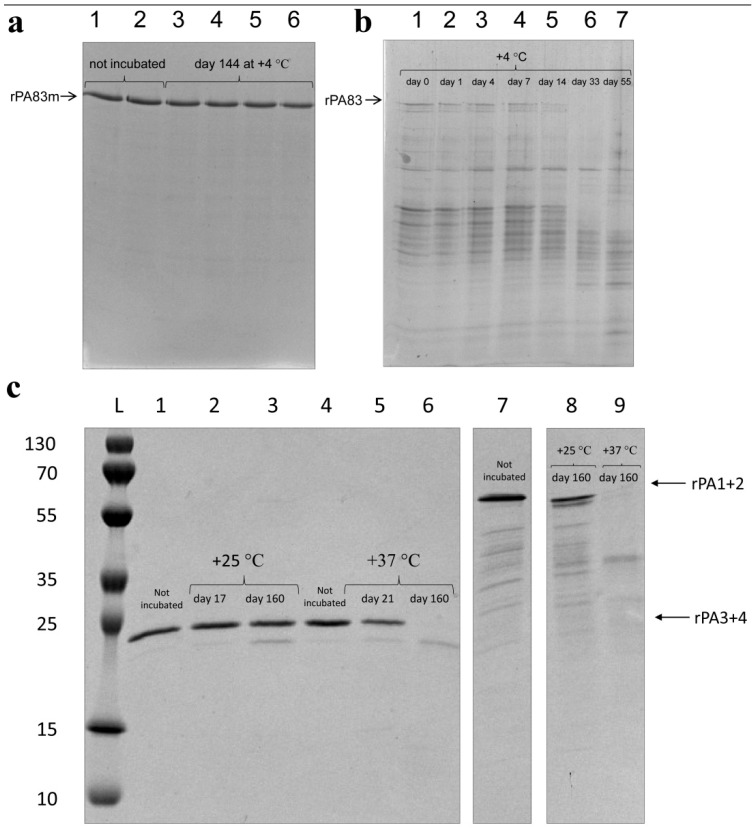
The stability of rPA83m (**a**), non-modified rPA83 (**b**), rPA3+4 and rPA1+2 (**c**); (**a**)—1, 2—rPA83m not exposed to incubation. 3–6—four independent replicates of a rPA83m sample incubated at +4 °C for 144 days. (**b**)—1—non-modified rPA83 not exposed to incubation, 2–7—non-modified rPA83 incubated at +4 °C for 1, 4, 7, 14, 33 and 55 days, respectively. (**c**)—L—protein molecular weight markers ladder (molecular weights, in kDa, are indicated on the left). 1, 4—rPA3+4 not exposed to incubation. 2, 3—rPA3+4 incubated at +25 °C for 17 and 160 days, respectively. 5, 6—rPA3+4 incubated at +37 °C for 21 and 160 days, respectively. 7—rPA1+2 not exposed to incubation. 8—rPA1+2 incubated at +25 °C for 160 days. 9—rPA1+2 incubated at +37 °C for 160 days. Original gels are presented in the Appendix A.

**Figure 5 pharmaceutics-14-00806-f005:**
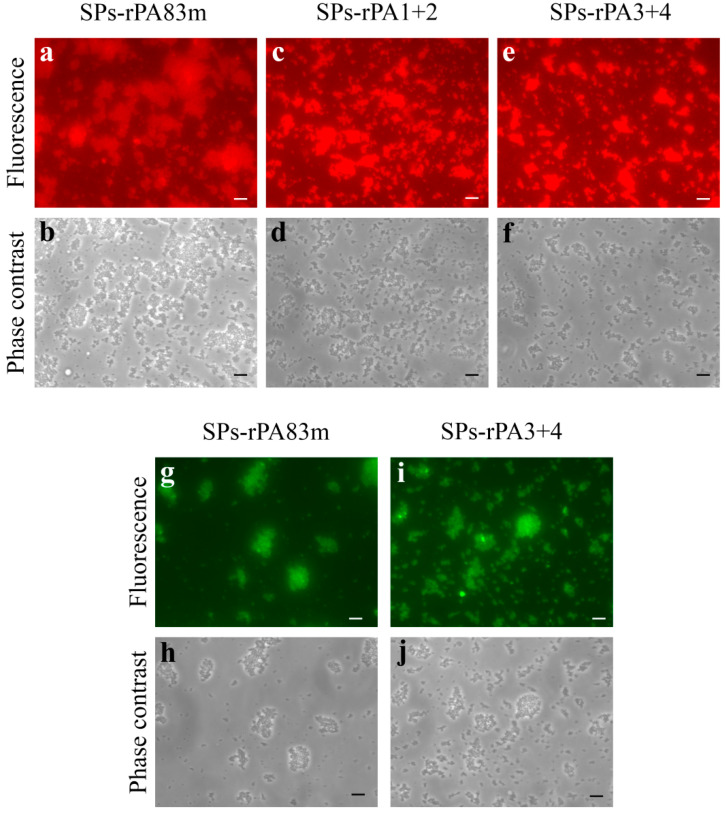
Compositions of SPs with anthrax modified recombinant antigens. Immunofluorescence analysis of SPs-rPA83m (**a**,**b**,**g**,**h**), SPs-rPA1+2 (**c**,**d**) and SPs-rPA3+4 (**e**,**f**,**i**,**j**) compositions. (**a**,**b**), (**c**,**d**), (**e**,**f**), (**g**,**h**), (**i**,**j**) in pairs are the same images presented in fluorescence and phase contrast modes, respectively. The SPs-protein compositions were treated with polyclonal antiserum to rPA83m (**a**,**b**), to rPA1+2 (**c**,**d**) or to rPA3+4 (**e**,**f**) or with chimeric monoclonal PA-neutralising antibodies (**g**–**j**) and secondary antibodies conjugated to Alexa Fluor^®^ 546 (**a**–**f**) or to CF™ 488A (**g**–**j**). Scale bars, 5 μm. SPs/antigen mass ratio within compositions is 10/1. Compositions were obtained in 1× PBS. Images of corresponding negative control samples (not incubated with primary antibodies) are presented in the Appendix A, respectively. Images of SPs-rPA1+2 compositions treated with chimeric monoclonal PA-neutralising antibodies and secondary antibodies conjugated to CF™ 488A as a negative control represented in Appendix A (k—fluorescence analysis, l—phase contrast).

**Figure 6 pharmaceutics-14-00806-f006:**
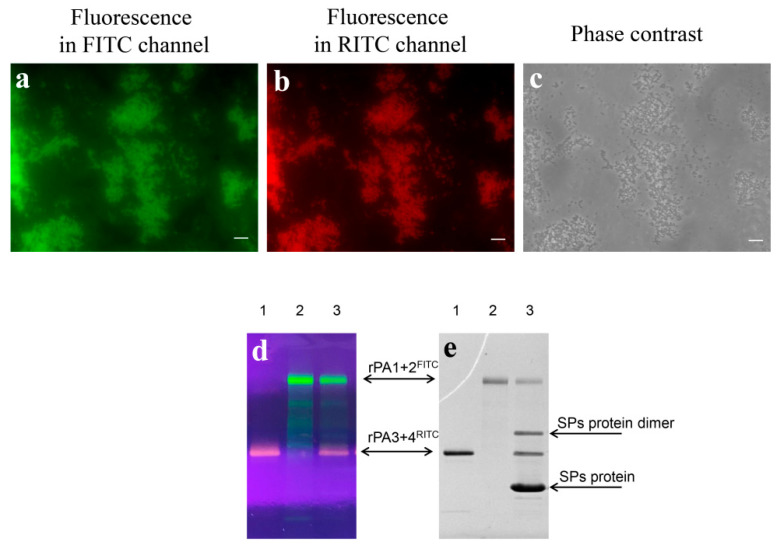
Simultaneous adsorption of rPA1+2 and rPA3+4 to SPs. (**a**–**c**)—Fluorescence microscopy of rPA1+2 labelled with fluorescein isothiocyanate (rPA1+2^FITC^) and rPA3+4 labelled with rhodamine isothiocyanate (rPA3+4^RITC^) simultaneously adsorbed to SPs. Images are presented in FITC fluorescence channel (**a**), RITC fluorescence channel (**b**) and phase contrast mode (**c**). Images of control samples (SPs compositions with rPA1+2^FITC^ or rPA3+4^RITC^ separately and uncoated SPs) fluorescence analysis are presented in the Appendix A. (**d**,**e**)—Electrophoresis analysis of modified recombinant anthrax antigens labelled with fluorophores in 8–20% SDS-PAGE. 1—rPA3+4^RITC^; 2—rPA1+2^FITC^; 3—Compositions of rPA3+4^RITC^ and rPA1+2^FITC^ with SPs. Unstained gel under UV-light (**d**) and Coomassie blue G-250 staining (**e**). The SPs/rPA1+2^FITC^/rPA 3+4^RITC^ mass ratio within Compositions was 20/1/1. Compositions were obtained in 1xPBS. Original gel is presented in the Appendix A.

**Figure 7 pharmaceutics-14-00806-f007:**
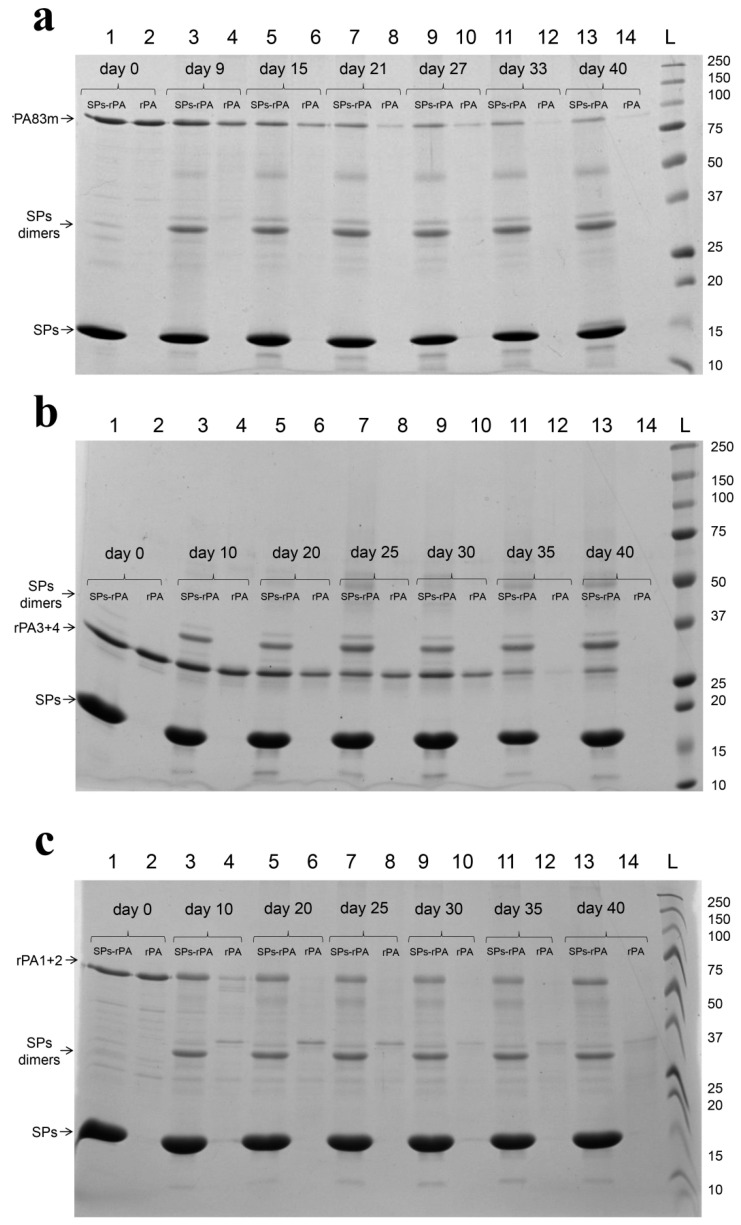
Comparative stability of SPs-antigen compositions and individual antigens at +37 °C for rPA83m (**a**), rPA3+4 (**b**) and rPA1+2 (**c**). (**a**)—1, 3, 5, 7, 9, 11, 13—rPA83m within SPs-rPA83m compositions incubated at +37 °C for 0, 9, 15, 21, 27, 33 and 40 days, respectively. 2, 4, 6, 8, 10, 12, 14—Individual rPA83m incubated at +37 °C for 0, 9, 15, 21, 27, 33 and 40 days, respectively. L—Protein molecular weight markers ladder (molecular weights, in kDa, are indicated on the right). (**b**)—1, 3, 5, 7, 9, 11, 13—rPA3+4 within SPs-rPA3+4 compositions incubated at +37 °C for 0, 10, 20, 25, 30, 35 and 40 days, respectively. 2, 4, 6, 8, 10, 12, 14—Individual rPA3+4 incubated at +37 °C for 0, 10, 20, 25, 30, 35 and 40 days, respectively. L—Protein molecular weight markers ladder (molecular weights, in kDa, are indicated on the right). (**c**)—1, 3, 5, 7, 9, 11, 13—rPA1+2 within SPs-rPA1+2 compositions incubated at +37 °C for 0, 10, 20, 25, 30, 35 and 40 days, respectively. 2, 4, 6, 8, 10, 12, 14—Individual rPA1+2 incubated at +37 °C for 0, 10, 20, 25, 30, 35 and 40 days, respectively. L—Protein molecular weight markers ladder (molecular weights, in kDa, are indicated on the right). Relative analysis of recombinant proteins degradation rate is presented in Table A1.

**Figure 8 pharmaceutics-14-00806-f008:**
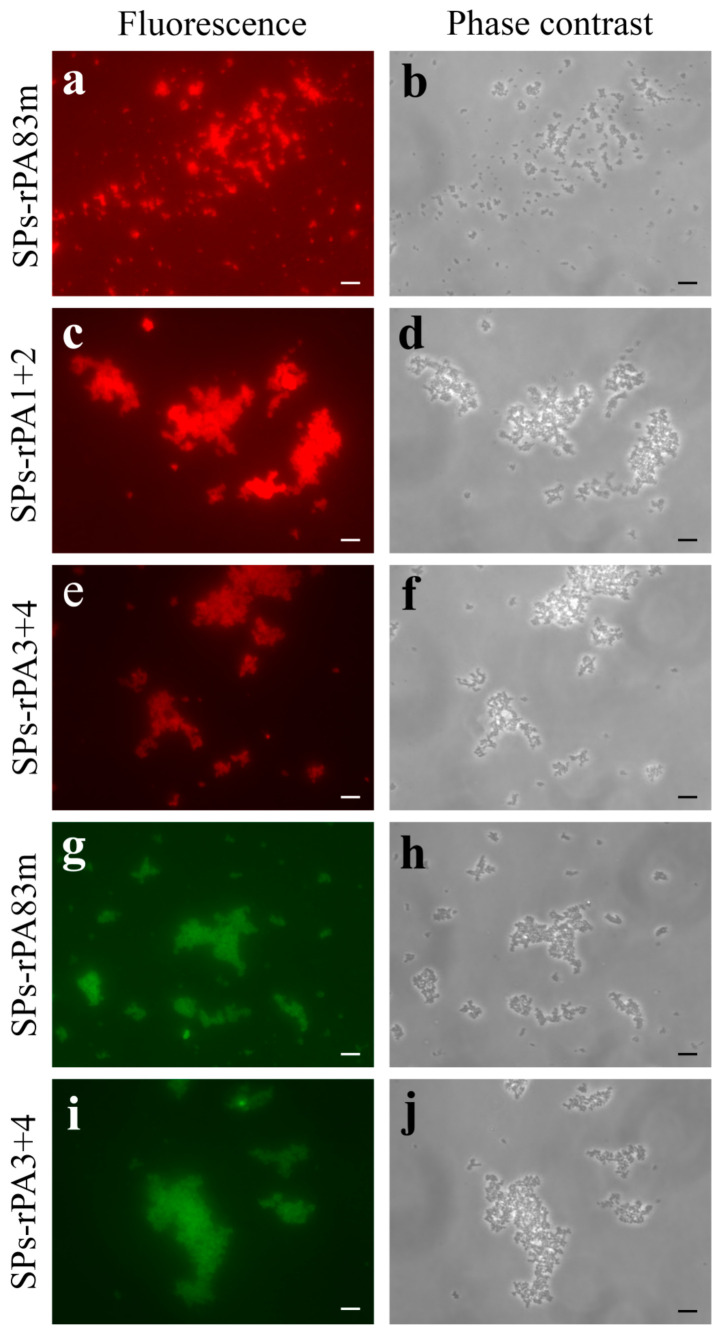
Antigenic specificity analysis of modified recombinant anthrax antigens within compositions with SPs after 40 days of incubation at +37 °C. Immunofluorescence analysis of SPs-rPA83m (**a**,**b**,**g**,**h**), SPs-rPA1+2 (**c**,**d**) and SPs-rPA3+4 (**e**,**f**,**i**,**j**) compositions. (**a**,**b**), (**c**,**d**), (**e**,**f**), (**g**,**h**), (**i**,**j**) in pairs are the same images presented in fluorescence and phase contrast modes, respectively. The SPs-protein compositions were treated with polyclonal antiserum to rPA83m (**a**,**b**), to rPA1+2 (**c**,**d**) or to rPA3+4 (**e**,**f**) or with chimeric monoclonal PA-neutralising antibodies (**g**–**j**) and secondary antibodies conjugated to Alexa Fluor^®^ 546 (**a**–**f**) or to CF™ 488A (**g**–**j**). Scale bars, 5 μm. SPs/antigen mass ratio within compositions is 10/1. Compositions were obtained in 1xPBS and incubated at +37 °C for 40 days. Images of corresponding negative control samples (not incubated with primary antibodies) are presented in the Appendix A, respectively.

**Table 1 pharmaceutics-14-00806-t001:** Oligonucleotides used for construction of rPA83m synthetic gene.

Template	Primer Name	Primer Sequence
pQE-30-rPA1+2	SphIFwd	5′-acgcggaagtgcatgcgtct-3′
pQE-30-rPA1+2	rPA1+2Rev	5′-*ggtttcctgaatctgcggca*-3′
pQE-30-rPA3+4	rPA3+4Fwd	5′-*tgccgcagattcaggaaacc*accgctcgtatcatcttcaac-3′
pQE-30-rPA3+4	PstIRev	5′-ctggtactgcaggttaccgt-3′

Underlined sequences represent the SphI and PstI restriction sites employed for cloning. The 20 nucleotide sequence corresponding to the 3′-end of the rPA1+2 gene is included in both rPA1+2Rev and rPA3+4Fwd primers and indicated in italics.

## Data Availability

All the relevant data are provided in this paper and in Supporting Information files.

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
