# Peer review of "Designing Stable Bacillus anthracis Antigens with a View to Recombinant Anthrax Vaccine Development"

_pharmaceutics, 2022, doi:10.3390/pharmaceutics14040806_

Round 1

Reviewer 1 Report

Ryabchevskaya et al. present the design and production of stable anthrax antigens. The manuscript is generally well written and the results are presented in a thorough fashion. The conclusion of the paper supports the presented results. Overall, I suggest acceptance of the paper after revision. However, some points remain unclear.

The protein expression and purification section leaves some open questions. The purification describes an immobilized metal affinity chromatography capture step under fully denaturing conditions. This implies that the proteins are expressed as polyhistidine-tagged inclusion bodies, yet this is not stated anywhere. The section obtaining genetic constructs does not clearly indicate the presence of a tag and the full sequences of the constructs are not given. If the constructs are produced with a His-tag, will this tag be removed? The authors should discuss a strategy for large scale production of this vaccine, as a His-tag is a regulatory no-go to my knowledge. Since the production process presented is apparently based on inclusion body refolding, some information on refolding yield and native conformation of the resulting protein should be given. Since the immunomicroscopy shows binding of polyclonal antiserum, at least some native conformation can be assumed. Do the authors have any additional data to support the successful refolding and native conformation of the antigen? This should be discussed in the manuscript.

The investigation of stability was based on SDS-PAGE analysis and immunomicroscopy. Is there any additional data to quantify the stability of the antigen? When using SDS-PAGE for measurement of stability, it is possible that the denaturing conditions of the assay lead to resolubilization of aggregates.  Since aggregation is claimed an additional pathway of antigen loss in line 513, the SDS-PAGE assay is not a useful way of assessing this degradation pathway. Size exclusion chromatography could be used to investigate aggregation of the free antigen, but the method is not useful for the SP stabilized antigens. An ELISA using the polyclonal or monoclonal antibodies should be able to quantify the loss of native epitopes over time. Addition of such a study would elevate the current manuscript immensely.

Furthermore, since no long-term stability study past 144 days at 4 °C is presented, what is the expected and required stability of this antigen? The authors give no indication as to what long term stability for a vaccine candidate would be sufficient and also do not even approximate the long-term stability (at 4 °C) of the presented constructs. Is the stability presented in this manuscript sufficient for a vaccine candidate? Furthermore, no data on manufacturability of the presented constructs is given, i.e. no yields. While this is not strictly necessary at this point, the authors should at least address the potential of scaling up production of the presented constructs.

Point for point:

Line 9 and 29: "... (both) humans and mammals" should be "...mammals, including humans".

ln86: The term biodegradable is usually associated with decomposition in the environment through microorganisms. Consider using a different term, such as no bioaccumulation in humans.

ln76&88: In line 75-77 it is stated that IV PA domain has the highest number of protective epitopes. In the current work, as stated in line 88-90, I and II PA domains are chosen. The current introduction does not make it clear that an extension of the repertoire towards all PA domains is the goal. I suggest adding a statement that makes clear that while rPA3+4 is already very useful, since it contains the epitope rich IV domain, the ability to produce all 4 domains would be better. I assume that the current work's focus on stability alone is first step towards investigating the ability of these antigens to induce protective immunity. If this is true, then this should also be stated.

ln146:Why was strain SG13009 chosen?

also ln146: I am only familiar with the term over-night culture, but not night culture.

ln155: IMAC is used for purification, but no His-tag is mentioned in the manuscript. Is PA natively metal binding or was a terminal His-tag used? If so, this should be clearly stated in the material and methods section.

ln168&174: Consider writing 5,000 and 10,000 to improve readability of the dilutions.

ln325-327: Awkward formulation, consider revising.

ln335-337: Since the neutralizing IgG does not bind rPA1+2 in Western blot (Figure 3c) it would have been the perfect negative control to use in Figure 5 immunomicroscopy. I would suggest to add this experiment to the manuscript, since it would be of interest what the immunomicroscopy result of a non-binding case would be.

ln367&432&509: I would refrain from using the word proven. Consider using shown/showed or confirmed.

ln518: TNA is not defined.

Figure 4 stability: rPA83m stability is investigated at 4 °C and rPA1+2 and rPA3+4  are investigated at 25 °C and 37 °C. Is 144 days at 4 °C long enough for vaccine stability. If degradation of rPA1+2 and rPA3+4 is observed after 160 days at 37 °C, what does this mean for long-term stability at 4 °C of rPA83m? Will the vaccine be stable for 2 years? Why is there still degradation? Is it possible to display the rPA1+2 data in panel 4c in a separate panel (i.e. 4d) as was done with rPA83m and rPA83?

SI system: There should be a space between numbers and units e.g. between 37 °C, not 37°C.

Author Response

We are grateful for the reviewer’s suggestions and comments that were helpful for improving the manuscript.

The immunostimulating activity and protectiveness of obtained proteins with existing conformations along with the effectiveness of implemented stabilization approaches will be examined in upcoming experiments on guinea pigs, which are a subject of our following researches. Here we performed the immunofluorescence analysis not only with polyclonal but also with monoclonal antibodies with PA-neutralizing activity. The interaction with PA-neutralising antibodies enables to believe that the conformation of obtained protein is acceptable to provide the eliciting neutralizing antibodies after immunization (the statement was added to the discussion section: lines 567-575). As far as concerned rPA aggregation that could be a reason to consider current stability evaluation by SDS-PAGE insufficient, we have assessed the protein aggregation state (Appendix C; discussion section: lines 543-549). When significant PA aggregation levels occur, large formations tend to be visible in solutions (Belton and Miller, 2013). Such formations were not detectable in rPA83m samples incubated at various temperatures (Figure A3 a, b). Also, the heat treatment of PA leads to the appearance of protein aggregates that can be precipitated through centrifugation (Singh et al., 2004; Ganesan et al., 2012). In current research, the supernatant of rPA83m samples after centrifugation at 16100g for 20 min were shown to contain the amount of protein equal to non-centrifuged samples (Figure A3 c, d), which indicates the lack of considerable rPA83m aggregates.

All final vaccine formulation details including the presence of His-tag, along with manufacturability and large scale production strategy can be discussed only after the complete preclinical trials, while the current work is an exploratory study.

Belton, D.J.; Miller, A.F. Thermal aggregation of recombinant protective antigen: aggregate morphology and growth rate. J Biophys 2013, 2013. https://doi.org/10.1155/2013/751091

Singh, S.; Singh, A.; Aziz, M.A.; Waheed, S.M.; Bhat, R.; Bhatnagar, R. Thermal inactivation of protective antigen of Bacillus anthracis and its prevention by polyol osmolytes. Biochem Biophys Res Commun 2004, 322, 1029-1037. https://doi.org/10.1016/j.bbrc.2004.08.020

Ganesan, A.; Watkinson, A.; Moore, B.D. Biophysical characterization of thermal-induced precipitates of recombinant anthrax protective antigen: evidence for kinetically trapped unfolding domains in solid-state. Eur J Pharm Biopharm 2012, 82, 475–484. https://doi.org/10.1016/j.ejpb.2012.05.019

Responses to point for point comments:

Point 1: Line 9 and 29: "... (both) humans and mammals" should be "...mammals, including humans".

Response 1: The MS text was modified according to the reviewer’s comment.

Point 2: ln86: The term biodegradable is usually associated with decomposition in the environment through microorganisms. Consider using a different term, such as no bioaccumulation in humans.

Response 2: The term “biodegradation” is widely used in studies on the biomedical application of plant viruses and their derivatives and their effect on the organism of animals and humans (Kaiser et al., 2007; Yildiz et al., 2011; Bruckman et al.,2014; Lee et al., 2017; Eiben et al., 2019; Venkataraman et al., 2021). In this context, the term “bioaccumulation” is appropriate, but it cannot replace “biodegradation” in meaning. In this regard, we would prefer to leave this term in the MS text.

Lee, P. W., Shukla, S., Wallat, J. D., Danda, C., Steinmetz, N. F., Maia, J., & Pokorski, J. K. (2017). Biodegradable Viral Nanoparticle/Polymer Implants Prepared via Melt-Processing. ACS nano, 11(9), 8777–8789. https://doi.org/10.1021/acsnano.7b02786.

Bruckman, M. A., Randolph, L. N., VanMeter, A., Hern, S., Shoffstall, A. J., Taurog, R. E., & Steinmetz, N. F. (2014). Biodistribution, pharmacokinetics, and blood compatibility of native and PEGylated tobacco mosaic virus nano-rods and -spheres in mice. Virology, 449, 163–173. https://doi.org/10.1016/j.virol.2013.10.035.

Yildiz, I., Shukla, S., & Steinmetz, N. F. (2011). Applications of viral nanoparticles in medicine. Current opinion in biotechnology, 22(6), 901–908. https://doi.org/10.1016/j.copbio.2011.04.020.

Kaiser, C. R., Flenniken, M. L., Gillitzer, E., Harmsen, A. L., Harmsen, A. G., Jutila, M. A., Douglas, T., & Young, M. J. (2007). Biodistribution studies of protein cage nanoparticles demonstrate broad tissue distribution and rapid clearance in vivo. International journal of nanomedicine, 2(4), 715–733.

Venkataraman, S., Apka, P., Shoeb, E., Badar, U., & Hefferon, K. (2021). Plant Virus Nanoparticles for Anti-cancer Therapy. Frontiers in bioengineering and biotechnology, 9, 642794. https://doi.org/10.3389/fbioe.2021.642794.

Eiben, S., Koch, C., Altintoprak, K., Southan, A., Tovar, G., Laschat, S., Weiss, I. M., & Wege, C. (2019). Plant virus-based materials for biomedical applications: Trends and prospects. Advanced drug delivery reviews, 145, 96–118. https://doi.org/10.1016/j.addr.2018.08.011.

Point 3: ln76&88: In line 75-77 it is stated that IV PA domain has the highest number of protective epitopes. In the current work, as stated in line 88-90, I and II PA domains are chosen. The current introduction does not make it clear that an extension of the repertoire towards all PA domains is the goal. I suggest adding a statement that makes clear that while rPA3+4 is already very useful, since it contains the epitope rich IV domain, the ability to produce all 4 domains would be better. I assume that the current work's focus on stability alone is first step towards investigating the ability of these antigens to induce protective immunity. If this is true, then this should also be stated.

Response 3: The MS text was modified according to the reviewer’s comment. The statements were added to the introduction and conclusions sections (lines 88-90 and 618-620).

Point 4: ln146:Why was strain SG13009 chosen?

Response 4: SG13009 is one of the strains available for us and it has been demonstrating high levels of expression of each recombinant anthrax protein since the very first attempts.

Point 5: also ln146: I am only familiar with the term over-night culture, but not night culture.

Response 5: The mistake was corrected.

Point 6: ln155: IMAC is used for purification, but no His-tag is mentioned in the manuscript. Is PA natively metal binding or was a terminal His-tag used? If so, this should be clearly stated in the material and methods section.

Response 6: The MS text was modified. The information on His-tags in recombinant proteins was added to the MS materials and methods section (paragraphs 2.1 and 2.2).

Point 7: ln168&174: Consider writing 5,000 and 10,000 to improve readability of the dilutions.

Response 7: The MS text was modified according to the reviewer’s comment.

Point 8: ln325-327: Awkward formulation, consider revising.

Response 8: The MS text was modified. The formulation was changed.

Point 9: ln335-337: Since the neutralizing IgG does not bind rPA1+2 in Western blot (Figure 3c) it would have been the perfect negative control to use in Figure 5 immunomicroscopy. I would suggest to add this experiment to the manuscript, since it would be of interest what the immunomicroscopy result of a non-binding case would be.

Response 9: The negative control image of SPs-rPA1+2 compositions treated with monoclonal neutralizing antibodies was added to Supplementary figure S3 (k-l).

Point 10: ln367&432&509: I would refrain from using the word proven. Consider using shown/showed or confirmed.

Response 10: The MS text was modified according to the reviewer’s comment.

Point 11: ln518: TNA is not defined.

Response 11: The definition of TNA was added.

Point 12: Figure 4 stability: rPA83m stability is investigated at 4 °C and rPA1+2 and rPA3+4  are investigated at 25 °C and 37 °C. Is 144 days at 4 °C long enough for vaccine stability. If degradation of rPA1+2 and rPA3+4 is observed after 160 days at 37 °C, what does this mean for long-term stability at 4 °C of rPA83m? Will the vaccine be stable for 2 years? Why is there still degradation? Is it possible to display the rPA1+2 data in panel 4c in a separate panel (i.e. 4d) as was done with rPA83m and rPA83?

Response 12: The conclusions about the sufficiency of recombinant anthrax proteins stability will be made after the experiments on their immunogenicity and protectiveness. The stability experiments carried out at +37 °C were designed to model the rapid protein ageing. The results of these experiments might not correlate directly with those of experiments carried out at +4 °C. Thus, no clear predictions on protein stability at +4 °C can be made based on stability at +37 °C.

Presumably some level of degradation of rPA proteins remains presumably because of several minor deamidation sites still exist in proteins structure. We have performed the substitutions of the major deamidation and proteolysis sites that were shown to lead to an increase in PA stability while not suppressing its antigenic properties.

Since the parts of Figure 4c represent a single gel (which is presented uncropped in the supplementary Figure S2c) we would not like to provide them separately in the MS and would prefer to keep rPA1+2 data as a part of Figure 4c.

Point 13: SI system: There should be a space between numbers and units e.g. between 37 °C, not 37°C.

Response 13: The MS text and images were modified according to the reviewer’s comment.

Reviewer 2 Report

In this study, the authors report the development of a stable anthrax vaccine composed of modified recombinant protective antigen (rPA83m) and rPA fragments (rPA 1+2 and rPA 3+4) combined with tobacco mosaic virus-based spherical particles (SP).  The modified rPA and rPA fragments were altered to resist deamidation and proteolysis which are two processes responsible for the low stability of non-modified rPA.  With the addition of the SP as an adjuvant, rPA83m and rPA 1+2/ rPA 3+4 had increased stability for prolonged times and various temperatures.  Although the authors reported on rPA 3+4 previously, this is the first report on rPA 1+2 and rPA83m.

  1. In Figure 5, why were PA-neutralising antibodies not used for rPA 1+2.

Author Response

The authors are thankful to the reviewer for the useful remarks.

Point 1: In Figure 5, why were PA-neutralising antibodies not used for rPA 1+2.

Response 1: Since neutralizing epitope corresponding to monoclonal antibodies used in the experiment was shown to be located either in PA domain III or PA domain IV, we have not initially provided the image of SPs-rPA1+2 compositions treated with monoclonal antibodies in Figure 5. According to reviewers’ recommendations, we added it to supplementary figure S3 (k-l) as a negative control.

Reviewer 3 Report

In this article, the authors have taken on the endeavor to increase the stability of the rPA protective antigens in the recombinant vaccines for Anthrax and thus pave the path to development of significantly more stable vaccine formulae out of E. coli. This is such a noble crusade as cell-culture platforms are way too expensive and as a result instability in the formulae of the current recombinant vaccines merits the scientific efforts.

The authors have succeeded in achieving this ambition with flying colors as the stability experiments of the modified rPA antigens have shown very promising enhancement; from total degradation of the non-modified rPA83 in the negative control to full stability after 144 in the modified one. This finding; as the crown jewel of the findings has been very well depicted in results and also well-discussed in the discussion. Therefore; supplemented by novelty, these findings grants the manuscript scientific merit an possible further industrial development.

Moreover, the entire spectrum of the experiments were simple and yet rich, have been carried out with perfect prejudice and are fully depicted and discussed. The immunofluorescence and SDS-PAGE analyses are all vividly well-shaped; showing the experience and prowess of the authoring scientists.

The manuscript has been very well drafted with almost no grammatical or illustration errors. The only graphical correction I can suggest is perhaps omitting the pictures of the sample Eppendorf tubes at the top of Figure A3 as they add no illustrative value to the figure.

After a final proofing and perhaps correcting Figure 3A, the manuscript is ready for publication in Pharmaceutics.

Author Response

We would like to thank the reviewer for the interest and evaluation of our paper.

Point 1: The only graphical correction I can suggest is perhaps omitting the pictures of the sample Eppendorf tubes at the top of Figure A3 as they add no illustrative value to the figure.

Response 1: When significant PA aggregation levels occur, large formations tend to be visible in solutions (Belton and Miller, 2013). The pictures of Eppendorf tubes presented in the Figure A3 demonstrate the lack of such aggregates in the solutions of obtained recombinant anthrax antigens. Thus, we would prefer to keep these pictures as a part of Figure A3.

Belton, D.J.; Miller, A.F. Thermal aggregation of recombinant protective antigen: aggregate morphology and growth rate. J Biophys 2013, 2013. https://doi.org/10.1155/2013/751091

Round 2

Reviewer 1 Report

The authors have addressed all points and I recommend publication of the revised manuscript.